# GROUNDED LANGUAGE LEARNING FAST AND SLOW

**Felix Hill, Olivier Tieleman, Tamara von Glehn, Nathaniel Wong, Hamza Merzic,**
**Stephen Clark**
DeepMind
London, UK
`{felixhill, tieleman, tamaravg, nathanielwong, hamzamerzic,`
`clarkstephen}@google.com`

## ABSTRACT

Recent work has shown that large text-based neural language models acquire a surprising propensity for one-shot learning. Here, we show that an agent situated in a simulated 3D world, and endowed with a novel dual-coding external memory, can exhibit similar one-shot word learning when trained with conventional RL algorithms. After a single introduction to a novel object via visual perception and language ("This is a dax"), the agent can manipulate the object as instructed ("Put the dax on the bed"), combining short-term, within-episode knowledge of the nonsense word with long-term lexical and motor knowledge. We find that, under certain training conditions and with a particular memory writing mechanism, the agent's one-shot word-object binding generalizes to novel exemplars within the same ShapeNet category, and is effective in settings with unfamiliar numbers of objects. We further show how dual-coding memory can be exploited as a signal for intrinsic motivation, stimulating the agent to seek names for objects that may be useful later. Together, the results demonstrate that deep neural networks can exploit meta-learning, episodic memory and an explicitly multi-modal environment to account for *fast-mapping*, a fundamental pillar of human cognitive development and a potentially transformative capacity for artificial agents.

## 1 INTRODUCTION

Language models that exhibit one- or few-shot learning are of growing interest in machine learning applications because they can adapt their knowledge to new information (Brown et al., 2020; Yin, 2020). One-shot language learning in the physical world is also of interest to developmental psychologists; *fast-mapping*, the ability to bind a new word to an unfamiliar object after a single exposure, is a much studied facet of child language learning (Carey & Bartlett, 1978). Our goal is to enable an embodied learning system to perform fast-mapping, and we take a step towards this goal by developing an embodied agent situated in a 3D game environment that can learn the names of entirely unfamiliar objects in a single exposure, and immediately apply this knowledge to carry out instructions based on those objects. The agent observes the world via active perception of raw pixels, and learns to respond to linguistic stimuli by executing sequences of motor actions. It is trained by a combination of conventional RL and predictive (semi-supervised) learning.

We find that an agent architecture consisting of standard neural network components is sufficient to follow language instructions whose meaning is preserved across episodes. However, learning to fast-map novel names to novel objects in a single episode relies on semi-supervised prediction mechanisms and a novel form of external memory, inspired by the dual-coding theory of knowledge representation (Paivio, 1969). With these components, an agent can exhibit both slow word learning and fast-mapping. Moreover, the agent exhibits an emergent propensity to integrate both fast-mapped and slowly acquired word meanings in a single episode, successfully executing instructions such as "put the dax in the box" that depend on both slow-learned ("put", "box") and fast-mapped ("dax") word meanings.

Via controlled generalization experiments, we find that the agent is reasonably robust to a degree of variation in the number of objects involved in a given fast-mapping task at test time. The agent also exhibits above-chance success when presented with the name for a particular object in the ShapeNet

arXiv:2009.01719v4 [cs.CL] 14 Oct 2020

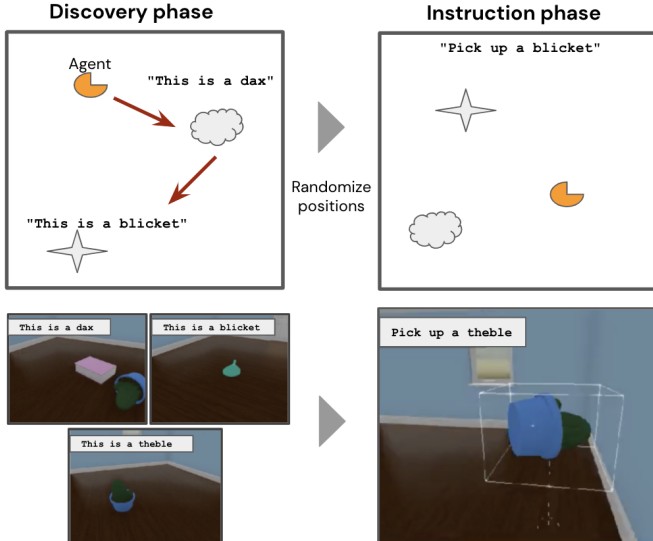

Figure 1: Top: The two phases of a fast-mapping episode. Bottom: Screenshots of the task from the agent's perspective at important moments (including the contents of the language channel).

taxonomy (Chang et al., 2015) and then instructed (using that name) to interact with a different exemplar from the same object class, and this propensity can be further enhanced by specific meta-training. We find that both the number of unique objects observed by the agent during training and the temporal aspect of its perceptual experience of those objects contribute critically to its ability to generalize, particularly its ability to execute fast-mapping with entirely novel objects. Finally, we show that a dual-coding memory schema can provide a more effective basis to derive a signal for intrinsic motivation than a more conventional (unimodal) memory.

## 2  AN ENVIRONMENT FOR FAST WORD LEARNING

We conduct experiments in a 3D room built with the Unity game engine. In a typical episode, the room contains a pre-specified number $N$ of everyday 3D rendered objects from a global set $G$. In all training and evaluation episodes, the initial positions of the objects and agent are randomized. The objects include everyday household items such as kitchenware (*cup, glass*), toys (*teddy bear, football*), homeware (*cushion, vase*), and so on.

Episodes consist of two phases: a *discovery* phase, followed by an *instruction* phase (see Figure 1).[1] In the discovery phase, the agent must explore the room and fixate on each of the objects in turn. When it fixates on an object, the environment returns a string with the name of the object (which is a nonsense word), for example "This is a dax" or "This is a blicket". Once the environment has returned the name of each of the objects (or if a time limit of 30s is reached), the positions of all the objects and the agent are re-randomized and the instruction phase begins. The environment then emits an instruction, for example "Pick up a dax" or "Pick up a blicket". To succeed, the agent must then lift up the specified object and hold it above 0.25m for 3 consecutive timesteps, at which point the episode ends, and a new episode begins with a discovery phase and a fresh sample of objects from the global set $G$. If the agent first lifts up an incorrect object, the episode also ends (so it is not possible to pick up more than one object in the instruction phase). To provide a signal for the agent to learn from, it receives a scalar reward of $1.0$ if it picks up the correct object in the instruction phase. In the default training setting, to encourage the necessary information-seeking behaviour, a smaller shaping reward of $0.1$ is provided for visiting each of the objects in the discovery phase.

Given this two-phase episode structure, two distinct learning challenges can be posed to the agent. In a slow-learning regime, the environment can assign the permanent name (e.g. "cup", "chair") to objects in the environment whenever they are sampled. By contrast, in the fast-mapping regime,

---

[1]Rendered images are higher resolution than those passed to the agent.

which is the principal focus of this work, the environment assigns a unique nonsense word to each of the objects in the room at random on a per-episode basis. The only way to consistently solve the task is to record the connections between words and objects in the discovery phase, and apply this (episode-specific) knowledge in the instruction phase to determine which object to pick up.

## 3 MEMORY ARCHITECTURES FOR AGENTS WITH VISION AND LANGUAGE

The agents that we consider build on a standard architecture for reinforcement learning in multi-modal (vision + language) environments (see e.g. (Chaplot et al., 2018; Hermann et al., 2017; Hill et al., 2020)). The visual input (raw pixels) is processed at every timestep by a convolutional network with residual connections (a ResNet). The language input is passed through an embedding lookup layer plus self-attention layer for processing. Finally, a *core memory* integrates the information from the two input sources over time. A fully-connected plus softmax layer maps the state of this core memory to a distribution over 46 actions, which are discretizations of a 9-DoF continuous agent avatar. A separate layer predicts a value function for computing a baseline for optimization according to the IMPALA algorithm (Espeholt et al., 2018).

We replicated previous studies by verifying that a baseline architecture with **LSTM core memory** (Hochreiter & Schmidhuber, 1997) could learn to follow language instructions when trained in the slow-learning regime. However, the failure of this architecture to reliably learn to perform above-chance in the fast-learning regime motivated investigation of architectures involving explicit external memory modules. Given the two observation channels from language and vision, there are various ways in which observations can be represented and retrieved in external memory.

**Differentiable Neural Computer (DNC)** In the DNC (Wayne et al., 2018), at each timestep $t$ a latent vector $\mathbf{e}_t = w(\mathbf{h}_{t-1}, \mathbf{r}_{t-1}, \mathbf{x}_t)$, computed from the previous hidden state $\mathbf{h}_{t-1}$ of the agent's core memory LSTM, the previous memory read-out $\mathbf{r}_{t-1}$, and the current inputs $\mathbf{x}_t$, is written to a slot-based external memory. In our setting, the input $\mathbf{x}_t$ is a simple concatenation $[\mathbf{v}_t, \mathbf{l}_t]$ of the output of the vision network and the embedding returned by the language network. Before writing to memory, the latent vector $\mathbf{e}_t$ is also passed to the core memory LSTM to produce the current state $\mathbf{h}_t$. The agent reads from memory by producing a query vector $q(\mathbf{h}_t)$ and read strength $\beta(\mathbf{h}_t)$, and computing the cosine similarity between the query and all embeddings currently stored in memory $\mathbf{e}_i$ ($i < t$). The external memory returns only the $k$ most similar entries in the memory (where $k$ is a hyperparameter), and corresponding scalar similarities. The returned embeddings are then aggregated into a single vector $\hat{\mathbf{r}}_t$ by normalizing the similarities and taking a weighted average of the embeddings. This reading procedure is performed simultaneously by $n$ independent read heads, and the results $[\hat{\mathbf{r}}_t^1, \ldots, \hat{\mathbf{r}}_t^n]$ are concatenated to form the current memory read-out $\mathbf{r}_t$. The vectors $\mathbf{e}_t$ and $\mathbf{h}_t$ are output to the policy and value networks.

**Dual-coding Episodic Memory (DCEM)** We propose an alternative external key-value memory architecture inspired by the Dual-Coding theory of human memory (Paivio, 1969). The key idea is to allow different modalities (language and vision) to determine either the keys (and queries) or the values. In the present work, because of the structure of the tasks we consider, we align the keys and queries with language and the values with vision. However, for different problems (such as those requiring language production) the converse alignment could be made, or a single memory system could implement both alignments.

In our implementation, at each timestep the agent writes the current linguistic observation embeddings $\mathbf{l}_t$ to the keys of the memory and the current visual embedding $\mathbf{v}_t$ to its values. To read from the memory, a query $q(\mathbf{v}_t, \mathbf{l}_t, \mathbf{h}_{t-1})$ is computed and compared to the keys by cosine similarity. The $k$ values whose keys are most similar to the query, $[\mathbf{m}^j]_{j \leq k}$, are returned together with similarities $[s^j]_{j \leq k}$. To aggregate the returned memories into a single vector $\mathbf{r}_t$, the similarities are first normalized into a distribution $\{\hat{s}^j\}$ and then applied to weight the memories $\hat{\mathbf{m}}^j = \hat{s}^j \mathbf{m}^j$. These $k$ weighted memories are then passed through a self-attention layer and summed elementwise to produce $\mathbf{r}_t$. As before this is repeated for $n$ read heads, and the results concatenated to form the current memory read-out $\mathbf{r}_t$. $\mathbf{r}_t$ is then concatenated with $\mathbf{h}_{t-1}$ and new inputs $\mathbf{x}_t$ to compute a latent vector $\mathbf{e}_t = w(\mathbf{h}_{t-1}, \mathbf{r}_t, \mathbf{x}_t)$, which is passed to the core memory LSTM to produce the subsequent state $\mathbf{h}_t$, and finally $\mathbf{e}_t$ and $\mathbf{h}_t$ are output to the policy and value networks.

| Architecture | Mean (S.D) accuracy $1e9$ training steps |
|---|---|
| LSTM | 0.33 (0.05) |
| LSTM + R | 0.61 (0.27) |
| DNC mem=1024 | 0.34 (0.01) |
| DNC mem=1024 + R | 0.64 (0.27) |
| TransformerXL mem=1024 | 0.32 (0.02) |
| TransformerXL mem=1024 + R | **0.98** (0.01) |
| DCEM mem=1024 | 0.33 (0.02) |
| DCEM mem=1024 + R | **0.98** (0.01) |
| TransformerXL mem=100 + R | 0.73 (0.35) |
| DCEM mem=100 + R | **0.98** (0.01) |
| Random object selection | 0.33 |

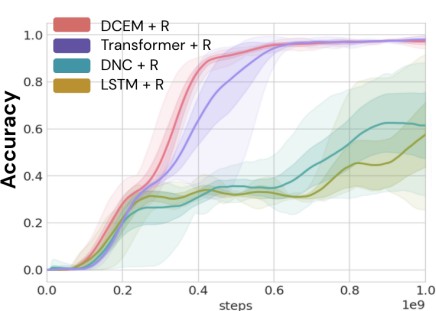

Table 1: Left: Performance when training on a three-object fast-mapping task with $|G| = 30$. *mem*: size of memory buffer/window *R*: with reconstruction loss. Right: Learning curves, each showing mean $\pm$ S.D. over 5 random seeds.

**Gated Transformer (XL)**    We also consider an architecture where the agent's core memory is a Transformer (Vaswani et al., 2017), including the gating mechanism from Parisotto et al. (2019). The only difference from Parisotto et al. (2019) is that we consider a multi-modal environment, where the observations $\mathbf{x}_t$ passed to the core memory are the concatenation of visual and language embeddings. We use a 4-layer Transformer with a principal embedding size of 256 (8 parallel heads with query, key and value size of 32 per layer). These parameters are chosen to give a comparable number of total learnable parameters to the DCEM architecture.

**Policy learning**    The agent's policy is trained by minimizing the standard V-trace off-policy actor-critic loss (Espeholt et al., 2018). Gradients flow through the policy layer and the core LSTM to the memory's query network and the embedding ResNet and self-attention language encoder. We also use a policy entropy loss as in (Mnih et al., 2016; Espeholt et al., 2018) to encourage random-action exploration. For more details and hyperparameters see Appendix A.4.

**Observation reconstruction**    In order to provide a stronger representation-shaping signal, we make use of a reconstruction loss in addition to the standard V-trace setup. The latent vector $\mathbf{e}_t$ is passed to a ResNet $g$ that is the transpose of the image encoder, and outputs a reconstruction of the image input $\mathbf{d}_t^{\text{im}} = g(\mathbf{e}_t)$. The image reconstruction loss is the cross entropy between the input and reconstructed images: $l_t^{\text{im}} = -\mathbf{x}_t^{\text{im}} \log \mathbf{d}_t^{\text{im}} - (1 - \mathbf{x}_t^{\text{im}}) \log(1 - \mathbf{d}_t^{\text{im}})$. The language decoder is a simple LSTM, which also takes the latent vector $\mathbf{e}_t$ as input and produces a sequence of output vectors that are projected and softmaxed into classifications over the vocabulary $\mathbf{d}_t^{\text{lang}}$. The loss is the cross entropy between the classification produced and the one-hot vocabulary indices of the input words: $l_t^{\text{lang}} = -\mathbf{x}_t^{\text{lang}} \log \mathbf{d}_t^{\text{lang}} - (1 - \mathbf{x}_t^{\text{lang}}) \log(1 - \mathbf{d}_t^{\text{lang}})$. For more details regarding the flow of information and gradients see Appendix A.4.

## 4    EXPERIMENTS

We compared the different memory architectures with and without semi-supervised reconstruction loss on a version of the fast-mapping task involving three objects ($N = 3$) sampled from a global set of 30 ($|G| = 30$). As shown in Table 1, only the DCEM and Transformer architectures reliably solve the task after $1 \times 10^9$ timesteps of training.

**DCEM vs. TransformerXL**    Importantly, the Transformer and DCEM are the two architectures that can exploit the principle of *dual-coding*. Since the inputs to the Transformer are the concatenation of visual and language codes, this model can recover the dual-coding aspect of the DCEM by learning self-attention weights $\mathbf{W}_k$ and $\mathbf{W}_q$ that project the language code to keys and queries, and weights $\mathbf{W}_v$ to project the visual code to values. Learning in the DCEM was marginally more sample-efficient, but this is perhaps expected given it was designed with fast-mapping tasks in mind. In light of this, is it really worth pursing memory systems with explicit episodic memories?

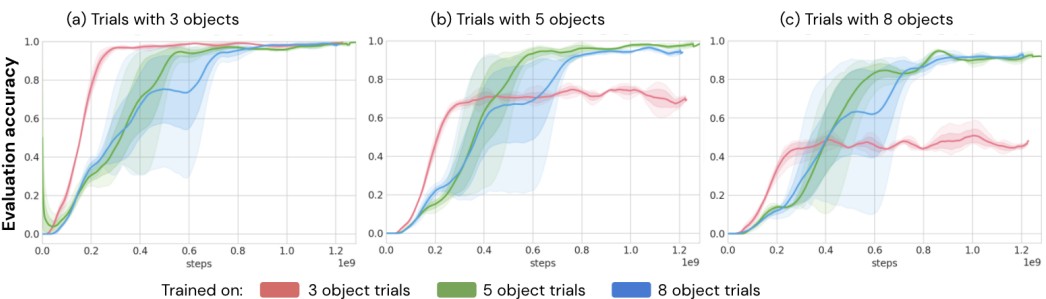

Figure 2: Accuracy of agents trained on probe trials involving a different number of total objects for agents meta-trained with different numbers of total objects.

To show one clear justification for external memory architectures, we conducted an additional comparison in which the memory windows of both the DCEM and the Transformer agents were limited to 100 timesteps (from 1024 in the original experiment), approximately the length of an episode if an agent is well-trained to the optimal policy. With a memory span of 100, the Transformer is forced to use the XL window-recurrence mechanism to pass information across context windows (Dai et al., 2019), while any capacity to retain episodic information beyond 100 timesteps in the DCEM must be managed by by the LSTM controller. In this setting we observed that the DCEM was substantially more effective (Table 1, left, bottom). While this imposed memory constraint may seem arbitrary, in real-world tasks working memory will always be at a premium. These results suggest that DCEM is more 'working-memory-efficient' than the Transformer agent. Indeed, by employing a simple heuristic by which the agent only writes to its external memory when the language observation changes from one timestep to the next, the DCEM agent with only 20 memory slots could solve the task with similar efficiency to a Transformer agent with a 1024-slot memory. See Appendix A.1 for these results and details of the selective writing heuristic.

## 4.1 GENERALIZATION

To explore the generalization capabilities of our agents, we subjected trained agents to various behavioural probes, and measured performance across thousands of episodes without updating their weights. Unless stated otherwise, all experiments in this section involve the DCEM+Recons agent.

**Number of objects** We first probed the robustness of the agent to fast-mapping episodes with different numbers of objects. In all conditions, the same objects appear in both the discovery and instruction phases of the episode, and the objects are sampled from the same global set $G$ ($|G| = 30$). As shown in Figures 2(b) and (c) (red curves), with the (default) meta-training setting involving three objects in each episode, performance on episodes involving five objects is approximately 70%, and with eight objects around 50%. This sub-optimal performance suggests that, with this meta-training regime, the agent does tend to overfit, to some degree, to the "three-ness" of its experience. Figure 2(b) shows, however, that the overfitting of the agent can be alleviated by increasing the number of objects during meta-training. Finally, Figure 2(a) confirms, perhaps unsurprisingly, that the agent has no problem generalizing to episodes with *fewer* objects than it was trained on.

**Novel objects** To probe the ability of the agents to quickly learn about any *arbitrary* new object, we instrumented trials with objects sampled from a global test set of novel objects $H : H \cap G = \emptyset, |H| = 10$. As shown in Figure 3, we found that an agent meta-trained on 20 objects (i.e. $|G| = 20$) was almost perfectly robust to novel objects. As may be expected, this robustness degraded to some degree with decreasing $|G|$, which is symptomatic of the agent specializing (and overfitting) to the particular features and distinctions of the objects in its environment. However, we only observed a substantial reduction in robustness to new objects when $|G|$ was reduced as low as three – i.e. a meta-training experience in which all episodes contain the same three objects (the first three elements of $G$ alphabetically, i.e. a *boat*, a *book* and a *bottle*).

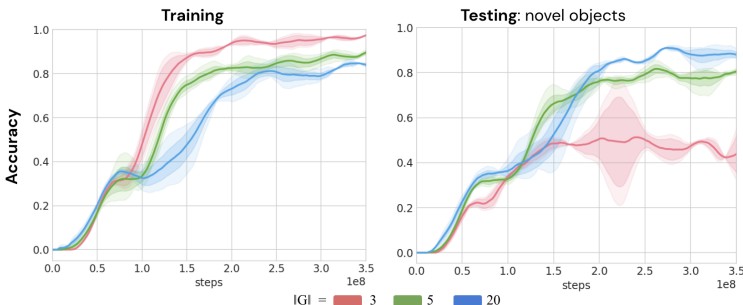

Figure 3: Accuracy during training and evaluation trials involving unfamiliar objects, for different sizes of global training set $G$. Curves show mean $\pm$ S.E. over 3 agent seeds in each condition.

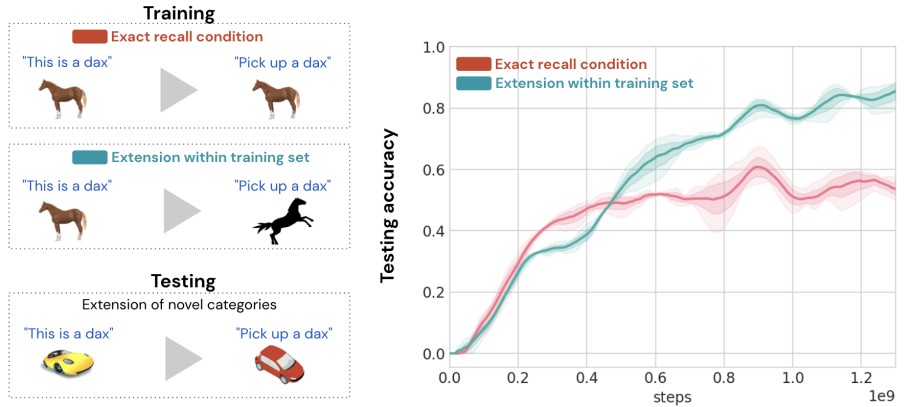

Figure 4: Accuracy of agents in fast-mapping trials requiring the extension of ShapeNet categories from a single exemplar. Curves show the mean $\pm$ S.E. over three agent seeds in each condition.

**Fast category extension** Children aged between three and four can acquire in one shot not only bindings between new words and specific unfamiliar objects, but also bindings between new words and *categories* (Behrend et al., 2001; Waxman & Booth, 2000; Vlach & Sandhofer, 2012). We conducted an analogous experiment by exploiting the category structure in ShapeNet (Chang et al., 2015). In a test trial, in the discovery phase the agent is presented with exemplars from three novel (held-out) ShapeNet categories (together with nonsense names). In the instruction phase, the agent must then pick up a *different and unseen* exemplar from one of these three new categories as instructed. As shown in Figure 4, when trained as described previously, the agent achieves around 55% accuracy on test trials, which is above chance (33%) but still a substantial error rate. However, this performance can be improved by requiring the agent to extend the training object categories as it learns. In this regime, three ShapeNet exemplars from distinct classes are encountered by the agent in the discovery phase of training episodes, and the instruction phase involves different exemplars from the same three classes. When trained in this way (which share similarities with *matching networks* (Vinyals et al., 2016)), performance on extending novel categories increases to 88%.

**Role of temporal aspect** Through ablations we found that both novel objects generalization and category extension relied on the agent reading multiple values from memory for each query. See A.2 for a discussion of these results, which suggest that the temporal aspect of the agent's experience (and learning from multiple views of the same object) is an important driver of generalization.

## 4.2 INTRINSIC MOTIVATION

The default version of the fast-mapping task includes a shaping reward to encourage the agent to visit all objects in the room. Without this reward, the credit assignment problem of a fast-mapping episode is too challenging. However, we found that the DCEM agent was able to solve the task

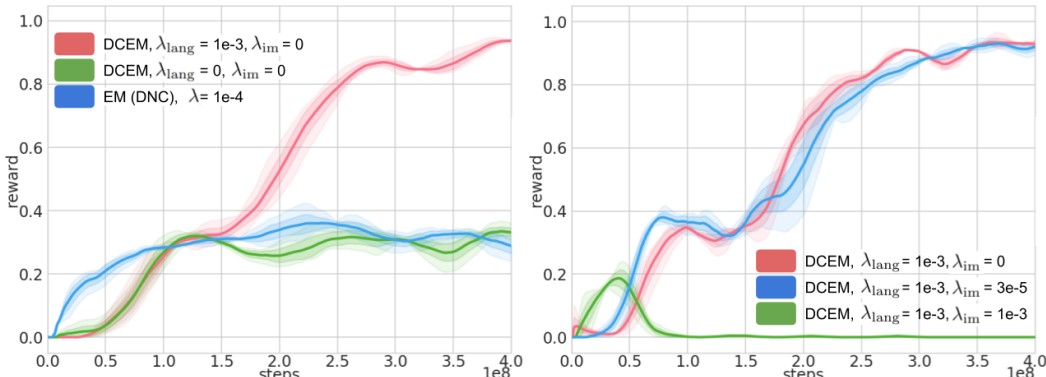

Figure 5: Accuracy of agents trained without shaping reward on the 3-object fast-mapping task with $|G| = 30$. Curves show mean $\pm$ S.E. across three seeds in each condition.

without shaping rewards by employing a memory-based algorithm for intrinsic motivation (NGU; Badia et al. (2020)). NGU computes a 'surprise' score for observations by computing its distance to other observations in the episodic memory, as described in Appendix A.4.3. The surprise score is applied as a reward signal $r^{\text{NGU}}$ which is added to the environment reward to encourage the agent to seek new experiences. We compared the effect of doing this in the DNC and the DCEM agents. For DCEM, the NGU computation can be applied to the memory's keys (language) column, its values (vision) column, or both. In the former case, the agent seeks novelty in the language space $r_{\text{lang}}^{\text{NGU}}$, and in the latter, in the visual space. The final reward is $r = r^{\text{ext}} + \lambda_{\text{lang}} r_{\text{lang}}^{\text{NGU}} + \lambda_{\text{im}} r_{\text{im}}^{\text{NGU}}$. As shown in Figure 5, we found that the DCEM agent (with $\lambda_{\text{lang}} = 10^{-3}$ and $\lambda_{\text{im}} = 3 \times 10^{-5}$) was able to solve the fast-mapping tasks without any shaping reward. This was not the case for the DNC agent, presumably because the required signal for 'language-novelty' is not approximated as well by the surprise score of the merged visual-language codes in the episodic memory.

### 4.3 INTEGRATING FAST AND SLOW LEARNING

To test whether our agents can integrate new information with existing lexical (and perceptual and motor) knowledge, we combined a fast-mapping task with a more conventional instruction-following task. In the discovery phase, the agent must explore to find the names of three unfamiliar objects, but in this case the room also contains a large box and a large bed, both of which are immovable. The positions of all objects and the agent are then re-randomized as before. In the instruction phase, the agent is then instructed to put one of the three movable objects (chosen at random) on either the bed or in the box (again chosen at random). As shown in Figure 6, if the training regime consisted of conventional lifting and putting tasks, together with a fast-mapping lifting task and a fast-mapping putting task, the agent learned to execute the evaluation trials with near-perfect accuracy. Notably, we also found that substantially-above-chance performance could be achieved on the evaluation trials without needing to train the agent on the evaluation task in any form. If we trained the agent on conventional lifting and putting tasks, and a fast-mapping task involving lifting only, the agent could recombine the knowledge acquired during this training to resolve the evaluation trials as a novel (zero-shot) task with less-than-perfect but substantially-above-chance accuracy.

### 4.4 RESULTS WITH ANOTHER ENVIRONMENT

To verify that the observed effects hold beyond our specific Unity environment, we added a new task to the DeepMind Lab suite (Beattie et al., 2016). Results for this task are given in Appendix A.3.

## 5 RELATED WORK

Meta-learning, of the sort observed in our agent, has been applied to train *matching networks*: image classifiers that can assign the correct label to a novel image, given a small support set of (image, label) pairs that includes the correct target label (Vinyals et al., 2016). Our work is also inspired by

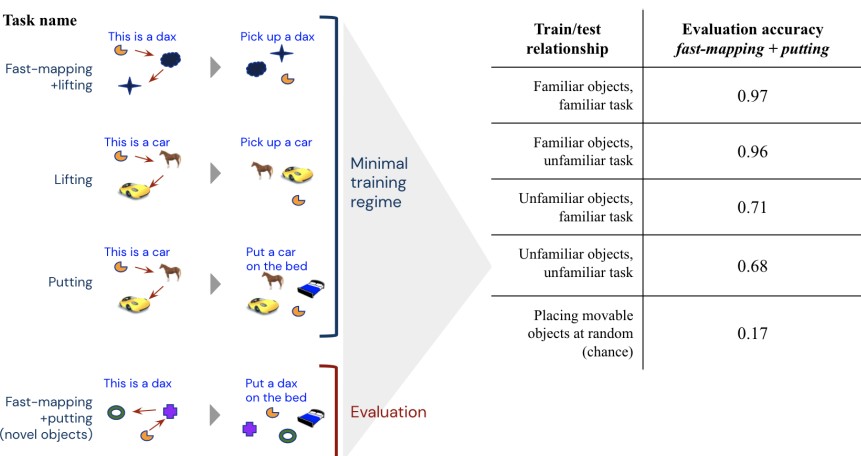

Figure 6: Right: The accuracy of the agent (accuracy $\pm S.E.$) on evaluation trials when exposed to different training regimes. Left: Schematic of the most impoverished training regime.

Snell et al. (2017), who propose a more efficient way to integrate a small support set of experience into a coherent space of image 'concepts' for improved fast learning, and Santoro et al. (2016), who show that the successful meta-training of image classifiers can benefit substantially from external memory architectures such as Memory Networks (Weston et al., 2014) or DNC (Graves et al., 2016).

In NLP, meta-learning has been used to train few-shot classifiers for various tasks (see Yin (2020) for a recent survey). Meta-learning has also previously been observed in reinforcement learning agents trained with conventional policy-gradient algorithms (Duan et al., 2016; Wang et al., 2019). In Model-Agnostic Meta Learning (Finn et al., 2017), models are (meta) trained to be easily tunable (by any gradient algorithm) given a small number of novel data points. When combined with policy-gradient algorithms, this technique yields fast learning on both 2D navigation and 3D locomotion tasks. In cognitive tasks where fast learning is not explicitly required, external memories have proven to help goal-directed agents (Fortunato et al., 2019), and can be particularly powerful when combined with an observation reconstruction loss (Wayne et al., 2018).

Recent work at the intersection of psychology and machine learning is also relevant in that it shows how the noisy, first-person perspective of a child can support the acquisition of robust visual categories in artificial neural networks (Bambach et al., 2018). When deep networks are trained on data recorded from children's head cameras, unsupervised or semi-supervised learning objectives can substantially improve the quality of the resulting representations (Orhan et al., 2020).

## 6 CONCLUSION

Our experiments have highlighted various benefits of having an explicitly multi-modal episodic memory system. First, mechanisms that allow the agent to query its memory in a modality-specific way (either within or across modalities) can better allow them to rapidly infer and exploit connections between perceptual experience and words, and therefore to realize *fast-mapping*, a notable aspect of human learning. Second, external (read-write) memories can achieve better performance for the same number of memory 'slots' than Transformer-based memories. This greater 'memory-efficiency' may be increasingly important as agents are applied to real-world tasks with very long episodic horizons. Third, in cases where it is useful to estimate the degree of novelty or "surprise" in the current state of the environment (for instance to derive a signal for *intrinsic motivation*), a more informative signal may be obtained by separately estimating novelty based on each modality and aggregating the result. Finally, an episodic memory system may ultimately be essential for fast knowledge *consolidation*. The potential for memory buffers and offline learning processes such as *experience replay* to support knowledge consolidation is not a new idea (McClelland et al., 1995; Mnih et al., 2016; Lillicrap et al., 2016; McClelland et al., 2020). For language learning agents, the need to both rapidly acquire *and retain* multi-modal knowledge may further motivate explicit

external memories. Retaining in memory visual experiences together with aligned (and hopefully pertinent) language (i.e. a dual-coding schema) may facilitate something akin to offline 'supervised' language learning. We leave this possibility for future investigations, which we will facilitate by releasing publicly the environments and tasks described in this paper.

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

## A    APPENDICES

### A.1    COMPARING TRANSFORMERXL TO DCEM WHEN MEMORY IS LIMITED

Both the TransformerXL and DCEM/DNC agents have a hyperparameter that determines the effective size of their explicit working memory. In a vanilla Transformer it determines the size of the window of timesteps that the network can be applied to for each forward (and backward) pass. To give models some chance of passing information beyond this hard constraint, TransformerXL architecture conditions each forward pass also on representations computed in the previous window, which establishes a form of recurrence over time from window to window. In our original experiments, this mechanism was not tested, because we set the window size to 1024 timesteps, which is longer than most episodes of the fast-mapping task, which are typically 80-120 timesteps for a well-trained agent.

To examine the performance of the TransformerXL in cases where it is required to pass information across context windows, we reduced the size of the window. For a fair comparison, we similarly reduced the equivalent parameter (the capacity in rows in the FIFO external memory) for the DCEM agent. As shown in Figure 7, for cases where the memory window size (or buffer) is reduced to (100) or below (50, 20) the normal episode length, the DCEM performs better than the TransformerXL agent. This suggests that the TransformerXL has difficulty making the necessary (visual and linguistic) information available to policy head when that information must be passed between context windows. Surprisingly, the DCEM was able to learn the tasks efficiently with a memory size of 50, which suggests that it must exploit its LSTM controller to retain sufficient information when its external memory begins to overflow. Both architectures fail when the memory size is reduced to 20, but in that case a DCEM agent can in fact learn the optimal policy if a simple heuristic for selective writing, described below, is employed. This highlights an advantage of explicitly read-write external memories; information can be managed via the reading or the writing process. It is not immediately obvious how the same strategy could be applied with a window-based memory architectecture like TransformerXL.

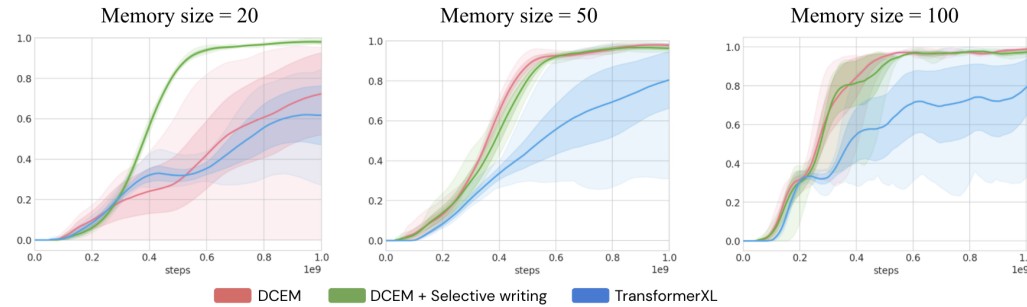

Figure 7:   Training success comparison between DCEM, DCEM with selective writing and TransformerXL for different sizes of memory-buffer (DCEM) or window (TransformerXL).

### A.1.1    SELECTIVE WRITING HEURISTIC

One advantage of explicit external (read-write) memories is that the flow of information to the agent's policy can be influenced by the writing process as well as the reading function. To verify this fact, we implemented a simple non-parametric heuristic writing condition in the DCEM architecture, whereby observations are written to the external memory when there is a change to the observation in the language channel. This heuristic aligns with the principle of dual-coding exploited elsewhere in the paper: while visual observations change continuously every timestep, changes to observed language are rare events that might signal some important change in the environment.

More formally, our heuristic relies on a window size parameter $w$ that we set to 3 in all cases. For observation $x_t = \{v_t, l_t\}$, a language change indicator $I_t \in \mathbb{N}$ is set as $I_0 = 0$

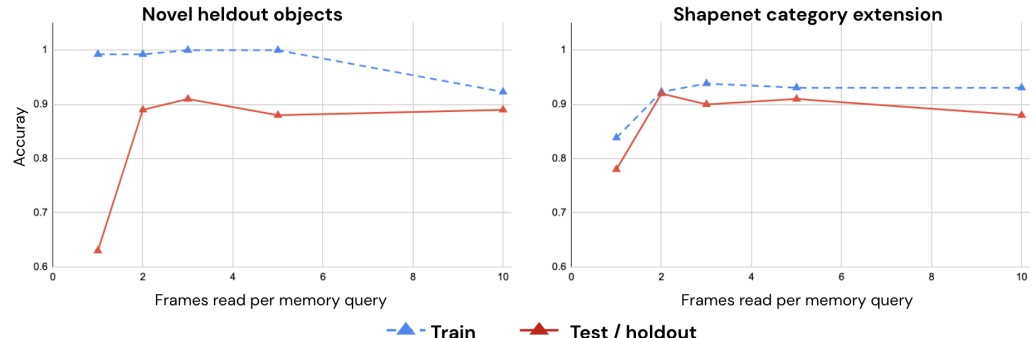

Figure 8: Training and test accuracy on two types of generalization tasks for agents that read different numbers of frames from their memory per query.

$$I_t = \begin{cases} t, & \text{if } l_t = l_{t-1} \\ I_{t-1}, & \text{otherwise.} \end{cases}$$

Then, the content $\mathbf{c}_t$ written to memory at $t$ is

$$\mathbf{c}_t = \begin{cases} \{\mathbf{v}_t, \mathbf{l}_t\}, & \text{if } t - I_t < w \\ \emptyset, & \text{otherwise,} \end{cases}$$

where, as before, $\mathbf{v}_t, \mathbf{l}_t$ are the agent embeddings of the visual and linguistic observations. Thus, memories are written for $w$ timesteps proceeding a change in the language observation.

## A.2 ROLE OF TEMPORAL ASPECT IN GENERALIZATION

In seeking to understand the mechanisms that support the generalization effects reported in the main paper, we found that the parameter $k$ was an important factor, where the top-$k$ memories are returned to the agent policy head per memory read. As the agent explores during the discovery phase of episodes, it writes multiple perspectives of the same object to memory. As shown in Figure 8, both its robustness to entirely novel objects (left) and its ability to extend categories from novel exemplars (right), as well as its ability to solve the training task, are enhanced when $k > 1$; i.e. when it determines which object to visit in the instruction phase *based on memories written from more than one view of each object*.

## A.3 VERIFICATION IN DEEPMIND LAB

At a high level, the design of an episode is very similar to the default fast-mapping task in the Unity environment. The agent must move down a corridor, bumping into (and collecting) three distinct objects. When an agent collects an object it is immediately presented with the (episode-specific) name for that object. After passing three objects, the corridor opens into a room containing two of the three objects found in the corridor. Upon entering the room, the agent is presented with the name corresponding to one of the two objects, and must bump into that object in order to receive a reward of $1$. As before, a shaping reward of $0.1$ is given as the agent collects each object in the corridor. Compared with the Unity environment, the DeepMind Lab action space is smaller (8 vs. 46 actions), the objects are larger, and the agent has no substantive way to interact with the objects (they disappear the moment the agent collides with them). Note also that the agent must choose between 2 (rather than 3) objects in the instruction phase, so an agent selecting objects at random would achieve 50% accuracy.

To provide some sense of the robustness and generality of the effects observed thus far, we applied the various agent architectures directly to this environment with no environment-specific tuning. As shown in Table 2, without any further tuning of the agent, we observe a similar pattern of results

| Architecture | Train. accuracy | Test (novel objects) |
|---|---|---|
| LSTM + R | 0.50 (0.02) | 0.40 (0.06) |
| DNC + R | 0.48 (0.04) | 0.30 (0.15) |
| TransformerXL mem=100 + R | 0.70 (0.28) | 0.55 (0.29) |
| DCEM mem=100 + R | 0.80 (0.26) | 0.65 (0.32) |
| Random object selection | 0.5 | 0.5 |

Table 2: Left: Architectures compared on DeepMind Lab after $5e8$ timesteps of training. Data show mean accuracy (S.D) across 5 seeds in each condition. *mem*: the agent's memory buffer size. *R*: with reconstruction loss. Right: Schematic of episode structure in the DeepMind Lab fast-binding tasks.

in DeepMind Lab as in the Unity room. As in that case, the Transformer and DCEM architectures performed best, with three and two seeds out of five (respectively) mastering the training task. As in the Unity environment, we also observed above-chance ability to apply fast-mapping knowledge zero-shot to unseen objects at test time.

## A.4 AGENT ARCHITECTURE DETAILS

### A.4.1 ARCHITECTURE DIAGRAMS

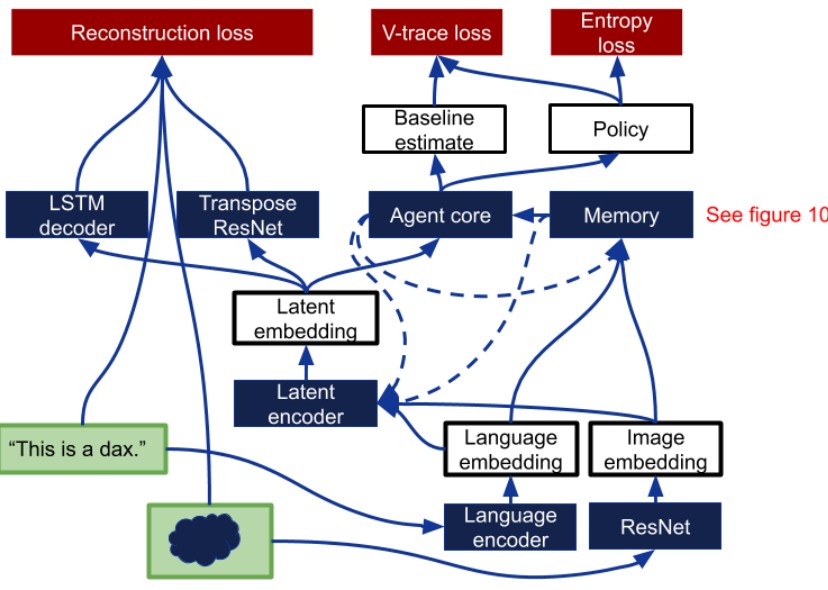

Figure 9: Agent architecture. See figure 10 for details of the dual coding episodic memory component. Dashed lines correspond to connections across timesteps.

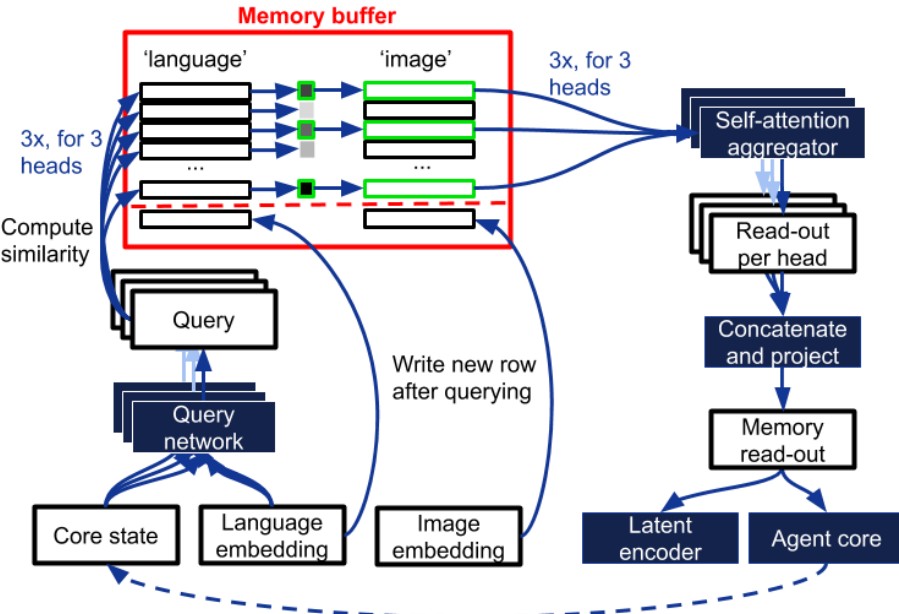

Figure 10: DCEM architecture. This corresponds to the 'memory' component in the agent architecture, Figure 9. NB: the similarity computation and selection of nearest neighbours is replicated for each read head, but not depicted here to avoid clutter. Dashed lines correspond to connections across timesteps.

| | |
|---|---|
| image width | 96 |
| image height | 72 |
| ResNet kernel size | $3 \times 3$ |
| convolutional layers per ResNet block | 2 |
| ResNet blocks | 2, 2, 2 |
| ResNet strides between blocks | 2, 2, 2 |
| ResNet number of channels | 16, 32, 32 |
| post-ResNet layer output size (visual embedding) | 256 |
| language encoder embedding size | 32 |
| language encoder self-attention key / query size | 16 |
| language encoder self-attention value size | 16 |
| language encoder output size (instruction embedding) | 32 |
| language decoder hidden size | 32 |
| number of memory read heads | 3 |
| memory aggregation self-attention key / query size | 256 |
| memory aggregation self-attention value size | 256 |
| latent representation size | 256 |
| core LSTM hidden size | 512 |
| policy latent size | 256 |
| value latent size | 256 |
| policy cost | 0.1 |
| entropy cost | $10^{-4}$ |
| reconstruction cost | 1.0 |
| return cost | 0.5 |
| discount factor | 0.95 |
| unroll length | 128 |
| batch size | 64 |
| Adam learning rate | $10^{-4}$ |
| Adam $\beta_1$ | 0 |
| Adam $\beta_2$ | 0.95 |
| Adam $\epsilon$ | $5 \times 10^{-8}$ |

Table 3: Agent hyperparameters (independent of specific architecture). The return cost (not discussed in the main text) is used to weight the baseline estimate term in the V-trace loss.

A.4.3   INTRINSIC MOTIVATION ALGORITHM

The intrinsic reward is based on the similarity (Euclidean distance) between the new embedding $\mathbf{e}$ and the nearest neighbors already present in memory $\{\mathbf{e}_i\}$. The average distance $\bar{\rho}$ used below is a lifetime average of all $\rho_i$ that is updated with every computation.

$$\rho_i = \frac{|\mathbf{e} - \mathbf{e}_i|^2}{\bar{\rho} + c}$$

$$k_i = \frac{\epsilon}{\max(\rho_i - \rho_{\min}, 0) + \epsilon}$$

$$s = \left(\sum_i k_i\right)^{1/2} + c$$

$$r_{\text{NGU}} = \begin{cases} 1/s & \text{if } s < s_{\max} \\ 0 & \text{otherwise} \end{cases}$$

The computation introduces the constants $c$, $\epsilon$, $\rho_{\min}$, and $s_{\max}$, and the number of neighbours $|\{\mathbf{e}_i\}|$ used for the similarity estimate. Table 4 lists the values we used.

| | | |
|---|---|---|
| number of nearest neighbours | $\|\{\mathbf{e}_i\}\|$ | 10 |
| smoothing constant for inverse distance / surprise | $c$ | $10^{-3}$ |
| similarity kernel smoothing constant | $\epsilon$ | $10^{-4}$ |
| cluster distance cut-off | $\rho_{\min}$ | $8 \times 10^{-3}$ |
| maximal similarity cut-off | $s_{\max}$ | 2.0 |

Table 4: Hyperparameters for NGU.

## A.5 ENVIRONMENT DETAILS

### A.5.1 UNITY ACTION SPACE

The following discrete actions (and strengths) are available to the agent in all experiments except for those in DeepMind Lab. The scalar strengths are translated into force and torque (for rotations) by the environment engine.

| Movement without grip | Fine grained movements without grip | Movement with grip |
|---|---|---|
| NOOP, | MOVE_RIGHT(0.05), | GRAB, |
| MOVE_FORWARD(1), | MOVE_RIGHT(-0.05), | GRAB + MOVE_FORWARD(1), |
| MOVE_FORWARD(-1), | LOOK_DOWN(0.03), | GRAB + MOVE_FORWARD(-1), |
| MOVE_RIGHT(1), | LOOK_DOWN(-0.03), | GRAB + MOVE_RIGHT(1), |
| MOVE_RIGHT(-1), | LOOK_RIGHT(0.2), | GRAB + MOVE_RIGHT(-1), |
| LOOK_RIGHT(1), | LOOK_RIGHT(-0.2), | GRAB + LOOK_RIGHT(1), |
| LOOK_RIGHT(-1), | LOOK_RIGHT(0.05), | GRAB + LOOK_RIGHT(-1), |
| LOOK_DOWN(1), | LOOK_RIGHT(-0.05), | GRAB + LOOK_DOWN(1), |
| LOOK_DOWN(-1), | | GRAB + LOOK_DOWN(-1), |

| Fine grained movments with grip | Object manipulation | Fine grained object manipulation |
|---|---|---|
| GRAB + MOVE_RIGHT(0.05), | GRAB + SPIN_RIGHT(1), | GRAB + PULL(0.5), |
| GRAB + MOVE_RIGHT(-0.05), | GRAB + SPIN_RIGHT(-1), | GRAB + PULL(-0.5), |
| GRAB + LOOK_DOWN(0.03), | GRAB + SPIN_UP(1), | PULL(0.5), |
| GRAB + LOOK_DOWN(-0.03), | GRAB + SPIN_UP(-1), | PULL(-0.5), |
| GRAB + LOOK_RIGHT(0.2), | GRAB + SPIN_FORWARD(1), | |
| GRAB + LOOK_RIGHT(-0.2), | GRAB + SPIN_FORWARD(-1), | |
| GRAB + LOOK_RIGHT(0.05), | GRAB + PULL(1), | |
| GRAB + LOOK_RIGHT(-0.05), | GRAB + PULL(-1), | |

### A.5.2 SHAPENET

ShapeNet contains 3D models of objects with a wide range of complexity and quality. To guarantee that the models are recognizable and of a high quality, we manually filtered the ShapeNet Sem dataset, selecting a subset of everyday semantic classes, ensuring that the selected models had a reasonable number of vertices, and reasonable size and weight dimensions. The selected classes (number of models in each class) were as follows, with a total of 1,437 models across 31 different classes:

*armoire (31), bag (11), bed (65), book (47), bookcase (13), bottle (26), box (37), bunk bed (9), chair (150), chest of drawers (133), coffee table (43), computer (6), floor lamp (68), glass (11), hammer (15), keyboard (5), lamp (100), loudspeaker (37), microwave (16), monitor (58), mug (12), piano (11), plant (31), printer (22), rug (36), soda can (20), sofa (145), stool (25), table (181), vase (66), wine bottle (7).*

