# OpenReview forum: "Grounded Language Learning Fast and Slow"
_ICLR.cc/2021/Conference — ICLR 2021 Spotlight_

### Official Review · AnonReviewer3 · 2020-10-22
**Well-written paper with an interesting contribution; scoping of the contribution claim is a bit wider than what experiments demonstrate [scope clarified via discussions and revisions]**

**Rating:** 8
**Confidence:** 3

**Review:**

This paper presents experiments for acquiring words via fast-mapping in an embodied environment. The technical contribution is interesting and solid, but the experiments fail to address some important questions that are yet scoped by the claims of the paper (namely, that learning is being done -both- fast and slow, as per the title). Notably, the paper is really well-written and readable, and the experiments on novel category + novel instance recognition are really convincing specifically for fast-mapping (4.1).

Experiments That Would Really Strengthen the Paper [I would love to hear your thoughts on these for rebuttal]:

- Fast mapping test where some object categories use the canonical name across trials, to demonstrate that the agent can still associate descriptive + instructive (not just instructive) langauge with both the fast and slow paradigm

- Experiments where the nonsense words are randomly generated characters in every episode of training and evaluation, to exactly tease out that the new word is being memorized and associated, not that the agent is memorizing that "dax" per episode changes meaning, which is itself knowledge about the word that renders it not truly "nonsense" or unseen. Currently, its meaning just becomes instead "a variable assigned to a new object per trial that I need to memorize".

Questions:

- Table 1 random object selection / LSTM perform at .33; does this mean there's a control module that decides how to do the navigation? Is the agent guided at test time to each object? This confuses me a lot! I thought it had to learn to go visit the objects and then to go get the one later, which seems unlikely to happen from an LSTM alone or random actions.


Improvements:

- I think it would be worth spending some time in the intro motivating why the agent is embodied. What does embodiment get us for the hypothesis that a network can be trained to do fast mapping? We can imagine just giving a sequence of images and associated descriptions (e.g., "this is a dax"), followed by the command "pick up the dax" together with all object images and the task of classification. What challenges does embodiment present that are missing from that paradigm? E.g., seeing the objects from non-canonical angles, having to find them, etc. There's some recent push in the NLP community in general for embodiment, so I think it's worth spending time on as motivation in this paper. [There's like a little bit? 4.1 mentions that learning from multiple views of the object is important, but this is sort of the entirety of what embodiment is buying us here versus a static image paradigm, so it's worth emphasizing, probably.]


- 4.3 is cool but doesn't -quite- do the fast-mapping test we'd like to see, which is that a word used consistently applies consistently. That would look more like including a few object categories whose names are used "correctly" across episodes, rather than one of the random names that are used inconsistently. This brings up a larger question of the consequences of using the same nonsense words for every trial, such that they can be memorized as being vacuous and in need of memorizing. Alternatively, the nonsense word could have been a random string to really test the memorization of an entirely novel token.



Nits:
- Figure 5 caption says these are accuracy curves, but the y axis is "reward"; is this task accuracy or the intrinsic reward values from surprisal?
- 4.4 why include this if there are no results in the paper for it? The constant references to the appendix are generally jarring, but this one is just sort of silly. It could be a single sentence maybe but not a whole subsection.

---

> ### Author Response · Authors · 2020-11-13
> **Thanks for constructive review. Some early responses and clarification.**
>
> Thank you for your review! We're grateful for the clear suggestions for improvement.
>
> Regarding "Fast mapping test where some object categories use the canonical name across trials" - we presume you mean that some small objects the agent has to lift are given permanent (canonical) names, rather than the bed or the storage tray that we introduce in 4.3. While conducting this research, we ran many such experiments exactly as you describe. One interesting observation was that, if an object is given a canonical name, we don't know by the agent's behaviour whether it learns that name in weights, or using the same activation+external memory strategy that is necessarily applied to the temporary object names. However, we can check this by including evaluation trials that don't include the discovery phase, only the instruction phase. On these truncated trials the agent can only score above chance if it has consolidated the name of objects into its weights. Interestingly, we find that an agent that is trained equally on canonical / permanent and temporary object bindings tends to apply the fast-mapping strategy to all object names (i.e. it can fail on the truncated evaluation trials with canonical names). Happily, this can be remedied by including some truncated trials with canonical names in the training regime. Once we do this, the agent learns to apply both slow-binding and fast-binding strategies together.
>
> The reason that we did not include this analysis in the paper itself (beyond the space limitations) was that we feel it is entirely superseded by the experiments in Section 4.3. There, not only does a single agent need to learn to do slow binding (to learn what a bed and what a storage tray are) and fast binding, but also needs to combine these two capabilities in a single episode to follow a single instruction. Thus, we think that the results in 4.3 already cover the suggested improvement to the paper that you request. Please let us know asap if this is not the case, and we'll try to remedy.
>
> Re. your questions. The LSTM condition is an agent whose core memory is an LSTM - in theory it could learn to do fast-binding just the same as any memory architecture, so it's not a priori guaranteed to fail on these tasks. However, we find that in practice it's very hard for the LSTM to retain a clear memory of a high dimensional input (video/images) necessary to execute fast binding. The random object selection here is a completely different thing - it's not a baseline agent that we ran. It is intended to show the expected performance of a hypothetical agent that selected objects in the instruction phase at random (since there are three in the room). I.e. to give readers a sense of what 'chance' performance is on the task, if chance is a policy that knows how to interact with objects but can't learn fast-mapping. A completely random agent would actually score much worse than 0.33, since it probably wouldn't ever pick anything up (as you noted, picking up requires a particular sequence of actions that is unlikely to happen by accident). We will make this much clearer to readers.
>
> We totally agree with your point about justifying the need for embodiment in this work. You noted that 4.1 is one place where embodiment makes a potential difference to the outcome compared to a supervised classifier. However, we feel that 4.3 is the section where embodiment is most fundamental in this work. Here, a verb-like motor policy (*putting*) combines with slow-learned words (*bed*, *tray*) and fast-bound objects such that the verb can apply to novel objects zero-shot. I.e. this example involves the integration of fast and slow word learning in a single example (as per the GPT3-examples), but also shows an agent learning to integrate fast-mapped knowledge in conjunction with a slow-learned motor behaviour. It is of course very speculative, but if in some hypothetical future we would like to teach an embodied agent the names of new objects, this example shows that the agent could integrate those objects with an existing repertoire of motor skills - i.e. usefully *do* things with those objects. We think this demonstration is a key contribution of this work, and we see no way in which it could be studied in a static (i.e. non-embodied) paradigm.
>
> Finally, regarding the nits (thanks) - what is plotted is extrinsic (environmental) reward, which is either 0 or 1 per episode, and therefore identical to accuracy (as there is no shaping reward). We will update the legend to make this clearer.  We will use the extra page to introduce 4.4 back into the main paper. Sorry if the appendices were hard to access; we submitted them as a separate file, but will add them to the main pdf to make referencing easier (other reviewers made the same observation).
>
> We will get back to you to let you know once the improvements to the paper that you propose have been added.

---

> > ### Comment · AnonReviewer3 · 2020-11-22
> > **Randomized names?**
> >
> > Thanks for these clarifications. This is really helpful. I think I buy the slow-mapping for receptacles, though of course there's the contraindicating factor that these are always static and not held objects.
> >
> > Any thoughts on randomized (e.g., always unseen) names so that the "dax"-style names don't become memorized variables for "this is something I need to fast-map"?

---

> > > ### Author Response · Authors · 2020-11-23
> > > **Synchronous fast+slow learning and generalisation to novel language input**
> > >
> > > Comment: Thanks for these clarifications. This is really helpful. I think I buy the slow-mapping for receptacles, though of course there's the contraindicating factor that these are always static and not held objects.
> > >
> > > Any thoughts on randomized (e.g., always unseen) names so that the "dax"-style names don't become memorized variables for "this is something I need to fast-map"?
> > >
> > > Thanks for getting back! I see the point about receptacles having different affordance in the environment also serving to distinguish them from other possible referents. Digging out our original experiments along these lines to provide some idea of the effects we observed - assuming equal 'dosage' of fast-binding (names randomized) and slow-binding (names held constant) training experience, and measuring performance on a slow-binding trials (i.e. using the same familiar names) with no discovery phase, the agent achieves ~69% accuracy (with chance at 33%) after 1G training steps, while its performance on fast-binding trials is over 95%. However, if we remove the discovery phase from the slow-binding training trials, the performance on both types of evaluation exceeds 90% simultaneously. We think this shows that these models can reasonably be 'taught' to apply slow or fast-binding memory strategies as the environment dictates. Because we (and you) think this tradeoff is interesting, we will include these slow-binding variants in the tasks in those that we release publicly.
> > >
> > > Regarding the 'always unseen' names, this is another v interesting question. Our focus here was on how the agent quickly infers and generalizes structure in its perception and binds that to word-symbols (which turned out to require a lot of innovations to get working, and yielded numerous interesting generalization effects). We expressly avoided the question of how an agent might cope with entirely unfamiliar character, morpheme, or 'sound'-complexes because, on initial consideration, we realised that it's a really fascinating research question that would warrant its own comprehensive study. Among the many questions that this work might investigate include: what sort of (frequency) distribution of characters/word-pieces/'sounds' does a model need to experience before it can reasonably interpret unfamiliar complexes? Does the model trained on language from a common distribution of text do better for certain unfamiliar word-like strings (those composed of common morphemes) than others? Would a grounded model whose language encoder was also trained auto-regressively on text have representations that better facilitated this sort of lexical or morphological generalization? We know from GPT3 that word-piece models trained on billions of words can do a decent job with held-out word-like strings, but is it possible to achieve this with less data? And on grounded tasks that require unfamiliar words to be both segmented and aligned with a perceptual world and an action policy?
> > >
> > > Following your raising of this it has become clear to us that others would also find these questions interesting, so we've added a short discussion in the final section raising this issue, delineating it as beyond the scope of the present work and setting out how future work might address some of these challenges.

---

> > > > ### Comment · AnonReviewer3 · 2020-11-24
> > > > **Thanks**
> > > >
> > > > Thanks for the additional details and discussion. With these changes and clarifications, it's clear to me that this paper is a strong contribution and should be accepted.

---

### Official Review · AnonReviewer1 · 2020-10-28
**Interesting results on grounded language learning and fast-mapping in a 3D world**

**Rating:** 8
**Confidence:** 4

**Review:**

# Overall review

The authors use a 3D world to explore grounded language learning, in which an agent uses RL to combine novel word-learning with stably acquired meanings to successfully identify and manipulate objects.  They show that a novel, psychologically-inspired memory mechanism is more memory-efficient than Transformers (both of which outperform plain LSTMs) and that it exhibits surprisingly robust generalization to novel action-object pairs.  The results should be of interest to many working in grounded language / multimodal representation learning, and the experiments are thorough and well-motivated.

Pros:
* Interesting environment for combining fast-mapping with stable language learning in a grounded task.
* Novel memory architecture, shown to improve memory-efficiency.
* Psychologically-motivated and thorough experimentation, demonstrating surprising level of generalization.

Cons:
* A figure describing the three memory architectures in addition to / instead of the plain text could have helped compare / contrast them.
* More analysis of how the agent uses its memory would be welcome.


# Minor comments

* 4.1: "Unless stated otherwise, all experiments in this section involve the DCEM+Recons agent".  I think this means with full memory size (1024) and not the smaller one (100), but it would help to clarify.

* Figure 2: while training with 3 objects and testing with more does not work, training with 5 and testing with 8 works as well as training with 8 and testing with 8.  What do the authors make of this?  Is there some kind of "threshold of diversity" beyond which the agent can generalize to more objects?  Such a threshold idea also seems consistent with the results in Figure 3.

* Fast category extension: these results show that if the agents are trained to pick up different exemplars of a category, they can do so in testing.  During training, there was one exemplar from each of three distinct categories.  I was curious if the authors experimented with relaxing that to environments where more than one exemplar of a category could be present, with the agent being rewarded for picking any of the correct exemplars.

* The environments the authors use seem like they would allow for testing of mutual exclusivity phenomena (see Ghandi and Lake 2020 [https://arxiv.org/abs/1906.10197] and references therein), by providing instructions with a novel word in a setting with one unseen object (or category).  I would be curious to see if their memory architecture does better with these phenomena than existing ones.


# Typographic comments

* p 3, "the current visual embedding l_t": l_t should be v_t

* Appendices were referenced in the text, but not included in the uploaded PDF (even though I believe this was allowed at ICLR).  These appendices seem very helpful for complete model/experiment details.


# Update

I thank the authors for their thoughtful reply and for incorporating the feedback.  The additional information is most welcome, and so I maintain my score of 8 for the paper.

---

> ### Author Response · Authors · 2020-11-15
> **Thank you for your review. We implement your fixes and respond to your questions.**
>
> Thank you for your review. We are glad that you consider the paper to be of merit. Thanks also for the suggestions for how to improve the manuscript. Regarding your questions and thoughts:
>
> - To provide some insight into how the agent uses its memory, we have introduced a figure to the supplementary material showing the observations corresponding to the embeddings retrieved from memory for an agent with num_memory_reads=5 for various different objects. This shows how the agent can make decisions about the referent of words based on memories of multiple viewpoints of an object, which results suggest contributes to the emergent capacity to fast-map unfamiliar objects.
>
> - We have added extra information in 4.1 (you were correct about the 1024 memory window) and the typo on page 3 (thank you)
>
> - We had added appendices as supplementary file, but in the revised version these are part of the main text to make consulting them easier
>
> - The idea about seeing multiple instances of a category during the discovery phase of an episode is very interesting. It is a bit reminiscent of the concept formation work of Xu and Tenembaum (2007). We'd certainly like to investigate this in future work, although to do so in a very controlled way may require adding to the environment objects whose properties we can alter automatically.
>
> - Similarly, we agree that it would be very interesting to apply mutual-exclusivity tests to the agent in future work, and we do intend to investigate this. By releasing the tasks we hope to enable others to study some of these questions too.

---

### Official Review · AnonReviewer4 · 2020-10-28
**Contains assortment of interesting results for grounded language learning with memory-based agents. However, results are more indicative than complete.**

**Rating:** 6
**Confidence:** 4

**Review:**

**Summary:** An agent following instructions in a grounded world is a core task in AI. This paper studies agent that accomplish this using memory-based architecture. This paper presents an argument for a multi-modal memory-architecture called DCEM whose key/queries and values are dependent on language and vision modalities respectively (or vice versa). An argument is made that this will be helpful for generalizing to novel language at test-time. Results are presented in a simple 3D domain containing several objects randomly sampled each time from a set of 30 objects. Task contain two types of instructions: "pick up an object" and "place an object on another object". Interaction proceeds in episodes where each episode contains a discovery phase where the agent learns the phrase associated with each object, and an instruction phase where the agent solves a given instruction. The proposed DCEM model outperforms baselines on various metrics and ablation. Importantly, it is shown that the DCEM can generalize to novel object names.

**Strengths:**
- The proposed architecture learns a near-optimal policy, requires few memory slots than baseline and can generalize to novel object names (fast mapping).
- It is argued that the language and observation key-value embedding can be used to compute a simple intrinsic reward that incentivizes reaching states (language+vision) whose distance in the embedding space is far from what has been encountered. Results show that this intrinsic reward removes the need for any reward shaping that is required by baselines. This claim is a bit more speculative since comparison with other exploration methods has not been provided. Distance in the observation space is also not a robust choice for defining intrinsic reward since similar observations can be actually very different in the latent state. E.g., consider observations which always contain a certain noise (say due to camera), then the intrinsic reward would never go down to 0 and the agent would keep exploring everywhere. In another example, walking down a long uniformly decorated corridor can result in a very small change in the observation but the agent would be making progress in the latent state. Such a path would not be adequately incentivized.

**Weakness:**
- The paper only investigates generalization of object names. This is kind of a low hanging fruit for generalization in language space. More difficult generalization include different textual styles, different choice of verbs, etc.
- DCEM architecture cannot generalize to more objects.
- Experiments are a bit underwhelming. The task is quite simple involving pick and place, with very simple language capabilities and a few objects. There is a plethora of complex environments now. E.g., see

1. ALFRED: A Benchmark for Interpreting Grounded Instructions for Everyday Task. Shridhar et al., 2020
2. Touchdown: Natural Language Navigation and Spatial Reasoning in Visual Street Environments. Chen et al., 2019
3. Vision-and-Language Navigation: Interpreting visually-grounded navigation instructions in real environments. Anderson et al., 2018.

I understand the advantage of control that comes with synthetic tasks, but replicating some results to more realistic tasks makes the claim stronger.

**Questions**
1. Is the language description changing with time? If not, wouldn't the language embedding l_t always be the same?
2. By "slot-based external memory" you mean memory with a fixed size of key-value pairs updated in a First-In-First-Out style? Section 3 can use some more explanation or an architecture Figure.
3. How do "independent read heads" work? Do they work similarly to "heads" in a transformer?
4. Did you try to generalize verbs? E.g., use a different word to refer to "Pick" during testing.
5. Does the background of the environment change across episodes (e.g., layout, colour of the wall, etc.)

---

> ### Author Response · Authors · 2020-11-13
> **Thank you for constructive review; we hope to work with you to address all minor concerns.**
>
> Thank you for your review! We're glad that overall you liked the paper.  We have a few questions that will help us to understand your concerns a bit better; we hope to work with you throughout this process to address these concerns and end up with the highest quality manuscript.
>
> We don't quite understand what you mean by  "DCEM architecture cannot generalize to more objects."?
>
> We showed in Figure 2 ways to train the DCEM agent so that it can generalize to more objects than it was trained on. We should note that this sort of generalization is very hard for humans; without a pen and paper handy or a lot of concentration we (the authors) struggle with episodes involving more than 3-4 objects. It is also tricky to extend beyond 8 objects without altering the setup completely, since the objects become very crowded together in the room, so the agent can't navigate between them. We're also not sure what a practical application would be for an agent that goes far beyond human capacity in this sort of temporary binding of object names.
>
> We'd love to investigate generalization to different textual styles and choices of verbs in future work. On what basis do you consider these to be more difficult than the different types of generalization we cover here (novel quantity of objects, novel object type, new category extension (4.1), compositional integration of tasks (4.3))? We'd love to know if there is some literature we are unaware of comparing the difficulty or importance of these types of generalization
>
> Thank you for the suggestion to use other environments in future work. While there are many fantastic environments out there that we considered for this work, the reason we chose a Unity environment is because of the physics, which made it possible to learn simplified versions of motor processes to interact with objects. For instance, to pick up an object and put it on the bed, our agent has to execute sequence of movements towards the object (until it is in range) then execute a sequence of object manipulation actions (something like GRAB+LOOK_UP repeated 4-5 times) then a sequence of movement actions to direct the agent to the correct location, then a sequence of object manipulation actions to place it down.
>
> We did not work with the environments that you suggest because they do not allow this sort of physical interaction with objects. Of those that you mention, only ALFRED has object interaction, and it is based on discrete binary actions (Pick_up, Open etc), so there isn't a sense in which an agent needs to learn a motor 'process' in order to learn how to apply a verb. More importantly, however, none of them include any tasks in which the agent needs to learn language quickly. They focus on (slow) learning of adult-like language instructions, whereas our focus is specifically on the challenge of fast-mapping word/object bindings (as children do) and integrating this knowledge into motor processes.
>
> Re. your questions:
> 1. The language changes during the discovery phase whenever the agent encounters a new object, and then it changes at the beginning of the instruction phase to instruct the agent. So, in a typical episode, it changes at least three times.
> 2. We are in the process of writing a clearer description of how the memory is populated, as also requested by other reviewers!
> 3. Yes, exactly; they come from query vectors produced by independent weight matrices. We will update the text to make this much clearer and let you know once that is done.
> 4. We can't quite understand the exact experiment that you have in mind here for generalization of verbs. Could you provide more details (or even just the research question that you're interested in?)
> 5. The layout and the colour of the walls does not change between episodes. The position and orientation of all objects and the agent are selected randomly per-episode, under the constraint that there is no collision between objects (so one is never on top of the other). We will make this detail more clear in the paper.

---

### Official Review · AnonReviewer2 · 2020-10-28

**Rating:** 8
**Confidence:** 3

**Review:**

This paper considers the problem of fast mapping: in a 3d environment, learning to explore an environment to learn a (new) mapping between objects and names, and then during a new phase, learning to pick up objects by their name ('pick up a dax'). This problem is challenging because the names change every time, so a model cannot simply learn connections between a shape and a static name (that persist longer than one episode).

Strengths:
* The paper defines the task and situates it within the context of a 3d environment, which could be useful for future work to build off of.
* The paper compares several sequence models for this task, and finds that a Transformer-XL and a a 'Dual-Coding Episodic Memory (DCEM)' model both do well. The DCEM model seems novel to this reviewer at least, and it seems to function like a hybrid LSTM and Transformer. It seems that the DCEM model performs better than the transformer-XL model when the number of timesteps is limited (so the transformer-Xl model needs to rely on its recurrence mechanism to propagate information).
* The paper tries several different generalization experiments involving more objects, new objects, a different evironment, and reward ablations. These generalization experiments provide us some insight as to what models learn in this setup.

Weaknesses:
At least to this reviewer, there are no major weaknesses (except perhaps that the dataset is a bit toy, which isn't a concern to me). However...
* It's not quite clear how much the memory bank is learning as opposed to the LSTM hidden state (which is also something mentioned in the supplemental). It would be interesting to learn a probe to measure when the model learns to assign an object to a name (if this is indeed something measurable by the memory bank alone).
* The reference to the 'DeepMind Lab Suite' is worded in a way that comes somewhat close to breaking anonymity...


-----

update: thanks for the clarification points + figure 9 :) I still recommend acceptance (score of 8).

---

> ### Author Response · Authors · 2020-11-15
> **Thank you for your review - a few points to clarify**
>
> Thank you for your kind review. We are glad that you found the work to have merit.
>
> Regarding your question about understanding the agent's use of memory, we have added a figure to the supplementary material (Figure 9) that we hope will add a little more clarity. It shows the five observations corresponding to visual embeddings read from the DCEM agent's memory at the first timestep of the instruction phase. The agent in question was configured to read (up to) five frames from its memory when making a query. We can see that the agent retrieves memories that are typically of the correct target object. These can then be directly compared to the current observation in order to decide on how to behave.
> To make the appendix easier to consult, we have added it to the main pdf document instead of in a separate supplementary file.
>
> Finally, the DeepMind lab suite is open source and anyone can add tasks. We have edited the text to make this clear.

---

### Author Response · Authors · 2020-11-16
**New version added, further improvements to come**

We have uploaded a version with the improvements made so far. A further set of improvements is to come.

---

### Decision · Program_Chairs · 2021-01-07
**Final Decision**

**Decision:**

Accept (Spotlight)

**Comment:**

The paper investigates the capacity for neural language models to perform fast-mapping word acquisition using a proposed multimodal external memory architecture. Much work exists that shows that neural models are capable of following instructions whose meaning persists across episodes (i.e., slow-learning), however much less attention has been paid to instruction-following in a one-shot learning context. Using a simulated 3D navigation/manipulation domain, the paper shows that the proposed multimodal memory network is capable of both slow and one-shot word learning when trained via standard RL.

The submission was reviewed by four knowledgable referees, who read the author feedback and engaged in discussion with the authors. The paper is topical---one-shot language learning for instruction-following using neural models is of significant interest of-late. The reviewers agree that the proposed multimodal memory architecture is both interesting and technically solid. The reviewers raised concerns about the experimental evaluation and the role of embodiment. The author feedback together with discussion with reviewers were helpful in resolving some of these issues. However, the authors are encouraged to ensure that the paper clearly motivates the importance of embodiment to slow learning and fast-mapping, particularly given the large body of work in language acquisition in robotics, a truly embodied domain, which is notably missing from the related work discussion.